# Review on High-Throughput Micro-Combinatorial Characterization of Binary and Ternary Layers towards Databases

**DOI:** 10.3390/ma16083005

**Published:** 2023-04-10

**Authors:** György Sáfrán, Péter Petrik, Noémi Szász, Dániel Olasz, Nguyen Quang Chinh, Miklós Serényi

**Affiliations:** 1Institute for Technical Physics and Materials Science, Centre for Energy Research, Konkoly-Thege út 29-33, 1121 Budapest, Hungary; 2Department of Materials Physics, Eötvös Loránd University, Pázmány Péter Sétány 1/A, 1117 Budapest, Hungary

**Keywords:** thin two- and three-component films, micro-combinatorial approach, microstructural, optical, mechanical properties, entire range of composition

## Abstract

The novel, single-sample concept combinatorial method, the so-called micro-combinatory technique, has been shown to be suitable for the high-throughput and complex characterization of multicomponent thin films over an entire composition range. This review focuses on recent results regarding the characteristics of different binary and ternary films prepared by direct current (DC) and radiofrequency (RF) sputtering using the micro-combinatorial technique. In addition to the 3 mm diameter TEM grid used for microstructural analysis, by scaling up the substrate size to 10 × 25 mm, this novel approach has allowed for a comprehensive study of the properties of the materials as a function of their composition, which has been determined via transmission electron microscopy (TEM), scanning electron microscopy (SEM), Rutherford backscattering spectrometry (RBS), X-ray diffraction analysis (XRD), atomic force microscopy (AFM), spectroscopic ellipsometry, and nanoindentation studies. Thanks to the micro-combinatory technique, the characterization of multicomponent layers can be studied in greater detail and efficiency than before, which is beneficial for both research and practical applications. In addition to new scientific advances, we will briefly explore the potential for innovation with respect to this new high-throughput concept, including the creation of two- and three-component thin film databases.

## 1. Introduction

Combinatorics, also known as combinatorial mathematics, concerns the problems of selection, arrangement, and operation within finite or discrete systems [1]. The combinatorial approach is ubiquitous, regardless of the involvement of mathematics. Indeed, the evolution of life on Earth is the greatest combinatorial experiment. Mother Nature conducts this experiment with millions of species and billions of individuals at a time through their confrontation with a changing environment, which, over millions of years, selects such individuals for survival and successful reproduction.

Even in our everyday lives, we use combinatorial approaches almost instinctively to organize and group our programs and tasks efficiently to save time, effort, and money.

Researchers can learn from nature and apply combinatorial solutions, preferably consciously instead of synthesizing and characterizing many individual samples. They can simultaneously use combinatorial techniques, collect or create groups of samples, and test them or have them interact with the environment to study their behavior more efficiently. Mother Nature’s infinite possibilities are a good fit with the trial-and-error approach, but due to humanity’s severely limited possibilities, researchers need to design their combinatorial experiments carefully.

One of the first combinatorial experiments for solving practical problems was published by Dorfman [2] during World War 2. He proposed a high-throughput method for screening syphilitic individuals from a large population of recruits. Since individual testing would have required great efforts and expenses and the incidence of the disease was only a few cases out of tens of thousands, he proposed group testing such that soldiers were divided into groups of one hundred individuals, whose blood samples were then pooled. The few pooled samples that presented the syphilis antigen were selected, and only persons from whom these samples originated were evaluated individually. This reduced the work and costs required by more than an order of magnitude. Dorfman’s example has clearly proven the benefits of combinatorial solutions.

In terms of materials science and technology, the structural, physical, and chemical properties of innovative materials depend largely on the proportion of their components. The traditional method for characterizing binaries and ternaries over a wide range of compositions is to synthesize and evaluate many samples, one for each composition, which is obviously inefficient. In contrast to individual studies, combinatorial techniques allow for the high-throughput characterization of compounds, thereby enabling the performance of comprehensive characterizations or even the creation of materials databases in a short time and with less effort.

The structure of this review is as follows: At the beginning of the paper, we briefly introduce typical combinatorial techniques, placing special emphasis on the characterization of thin films. This is followed by the main aim of the present work, that is, the presentation of the novel one-sample concept combinatorial technique, otherwise termed the micro-combinatory technique. This high-throughput technique will be demonstrated through examples based on results obtained from the comprehensive characterization of several two- and three-component systems.

## 2. Combinatorial Chemistry

The production of new materials and new compounds plays a major role in the development of chemistry, biology, and medicine. This largely depends on the emergence of new and more effective drugs, which requires the production of a large number of new organic compounds and the study of their effects. The development of a new drug requires the study and testing of tens of thousands of new compounds. Until the late 1980s, these compounds were both produced and tested for their effects individually. This situation was changed by the advent of combinatorial chemistry, which was first used in the pharmaceutical industry but now helps to produce not only drugs but also catalysts and many other molecules.

When using combinatorial chemistry, the aim is the rapid synthesis of molecules to find the most suitable compound; then, after filtering out the right compound, analogous, economical methods for synthesis are developed.

The first of the two main synthesis techniques used in combinatorial chemistry, based on Merrifield’s method [3], is the “Portioning-Mixing” (split-mix) synthesis method developed by Á. Furka [4,5,6]. The second one is called ”Parallel Synthesis”, which is based on the parallel performance of several traditional syntheses. This principle was first applied to the synthesis of peptides by Geysen and his colleagues [7]. Libraries created using the above or other methods should be tested. This justifies a third important technique in combinatorial chemistry, “High Throughput Screening”. The large number of available compounds has led to the emergence of novel approaches to the development of screening methods, usually consisting of parallel steps using robotics and sophisticated data-processing techniques [8]. Obviously, combinatorial chemistry experiments typically apply the multi-sample approach, which entails synthesizing numerous different samples in one experiment.

## 3. Combinatorial Characterization of Thin Films

Today’s integrated circuits, electronic devices, and sensors are mainly based on thin films with a wide range of physical and chemical properties. This is a strong motivation for researchers and technologists to focus on the efficient synthesis and characterization of thin film systems. Binary phase diagrams for solids are of paramount importance in science and technology, and as they have gained ground, there is an increasing need to create phase diagrams for thin films.

These techniques are becoming increasingly common in materials science. In 1965, K. Kennedy et al. developed a rapid combinatorial technique for the determination of ternary phase diagrams [9] using X-ray diffraction phase analysis for laterally resolved three-component films deposited from three offset e-beam evaporation sources onto a triangular substrate measuring 10 inches on side. The radiofrequency (RF) and direct current (DC) of laterally variable composition films from two separate targets [10,11] and concentrically arranged four-segment targets [12] were also reported.

The combinatorial technique proposed by Hanak in 1970 [13] along with the work of Priyadarshini et al. [14] represent major advances in the high-throughput synthesis of composition-spread thin films on few-centimeter-sized substrates. The setup of the latter is shown in Figure 1., in which the two components are deposited by thermal evaporation from two offset filaments.

The structural and compositional characterization of thin films relies primarily on microscopic techniques such as SEM and TEM. Depending on a laboratory’s equipment, such a sample can be examined with cutting-edge SEM microscopes equipped with an FEG electron source, which can provide direct insight into the microstructure of many PVD/sputtered coatings. However, for laboratories equipped with TEM or/and requiring atomic resolution or electron diffraction studies, it is indirect and inefficient to deposit a layer with a laterally varying composition on a substrate of a few centimeters and then prepare a series of TEM samples via a thinning procedure and use TEM to analyze the thinned samples individually. In addition, because the offset sources expose not only the near end but the opposite end of the substrate, the arrangement inhibits the deposition of pure materials, and combinatorial samples do not cover the entire composition range. The first combinatorial solutions directly developed for microscopic studies were published in the 1990s, and developments have been made with increasing interest, particularly using the “multi-sample concept”. Xiang and colleagues [15] used computer-controlled motion masking to synthesize numerous samples of different compositions at a density of 10,000 samples/in^2^. Roskov et al. [16] deposited polymer samples with different compositions onto individual TEM grids in one synthesis run. Rack et al. [17] applied DC magnetron sputtering from up to four offset targets to synthesize composition-spread layers over a substrate of several centimeters in size and predicted the local film composition via computer simulation. Julthongpiput et al. [18] prepared a series of organosilane specimens with a gradual transition from a hydrophobic to a hydrophilic nature that were investigated via atomic force microscopy and automated optical microscopy. Olk and Haddad [19] deposited 64 discrete Al_x_ Si_(1-x)_ samples in 1 run from up to 4 sputtering targets to control the microstructure of Al–Si alloys through variations in composition and growth temperature. Testing was conducted by atomic force microscopy and X-ray diffraction.

The combinatorial techniques described above are indeed effective for synthesizing variable samples on relatively large substrates or depositing numerous different samples in a single experiment. However, handling large numbers of distinct samples and examining them individually is inefficient, especially when using time-consuming, sophisticated analytical techniques such as transmission electron microscopy. The best solution would be a single transparent, composition-spread sample that covers the entire compositional range on a TEM grid suitable for direct microscopic examination.

A significant step forward in efficient analysis was taken by Barna’s group [20,21], who deposited binary layers of variable compositions on a single TEM grid. In their arrangement, a pair of offset thermal evaporation sources were used, exposing the grid through a 1 mm diameter aperture. The deposited film showed variable composition in the half-shadow behind the edge of the aperture. It was the first one-sample combinatorial solution that allowed for efficiency in both synthesis and characterization.

Figure 2 shows the image of the half-shadow area of a Cu–Ag sample deposited from two offset filaments (see Figure 2a) and the variation of the elemental composition of Cu and Ag vs. distance measured by EDS (see Figure 2b) across the area marked in Figure 2a.

However, despite its advantages, the method had significant shortcomings: the section with a variable concentration (half-shadow area) was very narrow, the concentration profile was difficult to reproduce, and the sample did not cover a full concentration range. These errors make it difficult to find a particular composition, clearly separate the phases, and collect the full data set, thus hindering the adoption of Barna’s method [20,21].

## 4. Methodology of the Novel Micro-Combinatorial Approach

To overcome the shortcomings of the methods described above, a one-sample concept combinatorial technique called the “micro-combinatory” (µ-combinatory) method was developed [22,23], which constitutes the topic of this review. The main goal of this method is to synthesize and characterize a single binary sample with a designed, laterally variable concentration profile on a TEM grid that represents the full compositional range. The micro-combinatorial technique, originally developed for TEM, has also proven to be applicable to further analytical measurements, such as EDS, XRD, XPS, RBS, ellipsometry, nanoindentation, etc. In this case, the technique only requires the sample size to be suitably scaled up to 1 × 0.5 inches. In this paper, the micro-combinatorial approach is presented in a non-exhaustive way, and its versatility is illustrated through a number of published examples.

The micro-combinatorial technique allows for the efficient synthesis of thin samples of laterally spread concentrations covering the entire compositional range, while satisfying the requirements of the given analytical technique. This may allow for the high-throughput characterization of an entire binary layer system. For TEM analysis, a thin film sample of components A and B, distributed in concentration, is deposited on a 3 mm grid, which is typically in the form of a strip 1 mm wide and 2 mm long. The strip is divided into three sections: a 0.25 mm long pure A section, a 1.5 mm long concentration gradient A–B section, and a 0.25 mm long pure B section. For further analysis, including with EDS, XRD, XPS, RBS, ellipsometry, and nanoindentation, the combinatorial sample is prepared on a Si, Ge, or silica substrate of, e.g., 25 × 12 mm^2^. Regardless of the size, this sample design allows for the production and analysis of all possible binary phases resulting from the combination of materials A and B along with the pure A and B components. 

Figure 3 depicts the experimental setup of the deposition and buildup of a micro-combinatorial TEM specimen. The sample is deposited in a single vacuum cycle by power-regulated dual DC magnetron sputtering on a TEM grid covered with carbon or SiO_x_ foil. Deposition is carried out through a narrow slot, which is moved by a stepping motor over the grid; meanwhile, the fluence of the magnetrons of targets A and B is regulated in sync with the slot movement. These steps are performed to synthesize the above-mentioned three sections of the combinatorial layer stripe. Details of the deposition process were published earlier in 2018 [22]. After deposition, the sample is removed from the vacuum system for structural, morphological, and energy-dispersive X-ray spectrometry (EDS) investigations via SEM and TEM.

## 5. Applications of the Micro-Combinatorial Technique in Materials Science

In the following brief overview, the versatility and effectiveness of the micro-combinatorial method is demonstrated through examples covering a wide range of material systems and characterization methods. Applications in advanced technologies render the study of thin films particularly important and timely, as they are in a non-equilibrium state in which they differ from the bulk material. Thin films contain a number of metastable, non-equilibrium, or unstable phases, the existence of which depends not only on their composition but also on the preparation conditions. Due to the large number of possible phases, there are no complete binary phase diagrams available for thin films (unlike the case for solids).

### 5.1. Al-Mn Thin Layer System

Upon heat treatment, bulk MnAl in the concentration range of 50–60 at%Mn undergoes a transformation from a hexagonal ε-MnAl phase to a metastable, ferromagnetic, tetragonal L1_0_ τ-MnAl phase [24,25]. Due to the high coercivity of the τ-phase, MnAl may be a suitable substitute for the Pt used in FePt and CoPt hard magnetic coatings, e.g., in recording media. However, different additional phases may exist in thin films, and the concentration range of the different phases known in the bulk state may differ in composition. This calls for a comprehensive study of the structure and morphology of the MnAl thin film system, for which the efficient micro-combinatorial method is well suited [26,27]. A variable concentration Mn-Al sample has been deposited at room temperature onto a Mo TEM grid covered with a SiOx support layer (see Figure 4).

The concentration gradient section at the center of the Mn-Al strip was 1.5 mm long and surrounded by 0.25 mm of pure Mn and pure Al. The concentration gradient under the experimental conditions was 0.067 at%/µm. After the deposition of the combinatorial sample, a thin SiOx protective layer was applied to prevent oxidation of the sample. The experimental conditions and details of the deposition process were published in [22]. The sample was characterized in a 200 kV Philips CM20 TEM by bright and dark field imaging, Selected Area Electron Diffraction (SAED), and Energy-Dispersive X-ray Spectrometry (EDS). Figure 4. illustrates that the morphology and structure of the AlMn binary layer system can be revealed over the entire composition range by measuring a single sample.

Figure 4A shows the photo of a MnAl combinatorial sample on a Mo TEM grid. The length of the arrow (1.5 mm) corresponds to the length of the gradient section. The elemental distribution along the sample measured by EDS is shown in Figure 4B. The Mn and Al concentration profiles are acceptably linear as a function of the position. However, at the boundaries between the one- and two-component regions, the profiles are curved because the width of the gap used to distribute the deposited layer is not zero. This is a minor disadvantage because the composition can be checked by EDS at any place as well as within the curved sections.

The TEM micrographs and SAED insets in Figure 5a–c show the typical morphology and structure of the µ-combinatorial sample at selected compositions. The films of 90 at%Mn in (a) present a fine-grained cubic Mn phase with some cubic MnO. This microstructure persists, while the Mn content decreases from 100% to about 50%. The 50–10%Mn concentration range is characterized by the presence of an amorphous MnAl phase, which is represented in (b) by the 40%Mn layer. Next to the amorphous phase, at 20%Mn and below, an fcc Al phase appears with a relatively large grain size. The grain size continues to increase, and the fcc Al phase becomes more prevalent as the Mn content decreases to 0%Mn.

Note that in addition to the phases of the layers deposited at room temperature, all the MnAl phases formed at an elevated temperature can be studied after the annealing of the deposited µ-combinatorial sample. For example, the 40%-Mn-containing film after annealing in a 500 °C/0.5 h Ar–H2 mixture (Figure 5d) shows a tetragonal Mn 14 Al 11 phase (Pdf 11-0520) identified by the appearance of 001, 110, and 200 rings in SAED. However, a detailed analysis of the heat-treated layers is outside the scope of this work. It should also be emphasized that both theoretical calculations and experimental results show that the diffusion effect between the adjacent film areas with different compositions is negligible.

Figure 6 shows the phases found in the as-deposited μ-combinatorial TEM sample via SAED intensity distribution plots as a function of an Al-Mn elemental composition. From the bottom to the top of the figure, the microstructural changes associated with decreasing Mn concentrations can be clearly followed (as could be seen in the selected examples in the previous Figure 5a–c). At 100%, 80%, and 60% manganese content, the sample is characterized by a fine-grained cubic α-Mn phase (Pdf 32-0637) with broad 222, 330, and 332 peaks and 111 and 220 peaks of cubic MnO (Pdf 75-0621) [28]. With 60%Mn content, the 330 peak becomes broader, i.e., more diffuse, which represents a continuous grain-size reduction. A further decrease in Mn concentration leads to amorphization. When the Mn content is reduced to 40%, an amorphous Mn(Al) solid solution is obtained. At the same time, the two peaks of the MnO phase disappear, presumably because the layer with a higher Al concentration is less sensitive to oxidation. Intensity diagrams for the 20%, 10%, and 0%Mn concentrations show a gradual decrease in the peak of the amorphous Mn(Al) solid solution and an increase in peaks identified with the 111, 200, and 220 reflections, showing the co-presence of an amorphous Mn(Al) phases and a crystalline fcc Al phase. The latter consists of Al grains of increasing size as the Al concentration increases. When the Mn concentration reaches 0%, i.e., 100%Al, the layer consists of an entirely fcc Al phase.

### 5.2. Al-Mg Films from Submicron Grain Size to Amorphous Structure

Due to their excellent mechanical properties, aluminum alloys in bulk form are indispensable materials in the automotive, construction, and aerospace industries and are, therefore, the focus of research [29,30].

The new micro-combinatorial method [31] has been used to effectively reveal the correlations between the structure and mechanical properties of thin Al-Mg films. This flexible method allows for the preparation of samples that meet the requirements of relevant measurement techniques. To characterize structure and morphology by TEM, a thin concentration gradient sample covering the entire compositional range was deposited on a TEM grid. To measure mechanical properties by nanoindentation [32], a series of 2 × 12 mm^2^ strips of selected discrete compositions were deposited on a single Si substrate in one experimental run to cover the technologically relevant composition range of an AlMg system (0–30 at%Mg) [31].

The TEM images in Figure 7a–d show the microstructures of the films at 0, 1, 10, and 30%Mg content, respectively. The fine grain size (~110 nm) of pure Al (0%Mg) shown in (a) decreases further when increasing the Mg concentration (~80 nm for 1%Mg), reaching ~40 nm for Al-10at%Mg (c). Finally, the layer appears dominantly amorphous at Al-30at%Mg (d). According to the SAED results, the layers exhibit an fcc Al(Mg) phase at each composition up to 30%Mg.

The nanoindentation measurements were performed on distinct strips prepared with selected compositions [31]. Figure 8a shows the indentation depth–load (h–F) curves of the AlMg layers at 0, 1, 5, 10, 20, and 30%Mg concentrations. It is evident that as low as a 1 at%Mg level of alloying leads to a remarkable decrease in the penetration depth of the indenter, i.e., a significant (~3×) hardness increase compared to pure Al. The nano-hardness (H) of the layers plotted against the Mg concentration in Figure 9 shows a further increase, reaching the remarkable high value of H_max_ = 8 GPa at an about 15 at%Mg concentration.

In the case of the Al-30%Mg film shown in Figure 8b, an interesting phenomenon, namely, a stepped behavior of the penetration curve, was observed. This is in complete agreement with both the dominant amorphous structure determined by TEM and SAED (Figure 7d) and with the appearance of deformation bands around the imprint as revealed by AFM (Figure 8b). Indeed, the discontinuous indentation curve is a phenomenon typical of the deformation of amorphous materials [33].

### 5.3. High Strength in AlCu Thin Films

Among their other advantageous mechanical properties, bulk AlCu alloys are also known for their special aging behavior [34], which has been well studied and documented [35]. However, in the form of thin films, this special alloy is hardly discussed and only in cases with low Cu concentrations [36,37]. To study the structural and mechanical properties of the AlCu thin film system in a wide compositional range, Draissia et al. [38,39] deposited several thin film samples using cathodic radiofrequency (RF) sputtering. They used composite targets in which a bulk Cu disk was inserted into a bulk Al ring, and by changing the surface fraction of the insert, the composition of the sputtered layer was adjusted. Though this method, 10 different, separate samples with Cu concentrations ranging from 1.8 to 92.5 at% were prepared [39]. The microstructural characteristics were determined by XRD and TEM, while the hardness of the layers was determined by nanoindentation.

Recently, the micro-combinatorial method has been applied to the structural and mechanical investigation of the AlCu thin film system [40], which has allowed for a more efficient and reproducible characterization over the whole concentration range compared to the traditional one-experiment–one-sample method. In a single vacuum cycle experiment using dual DC magnetron sputtering, 15 distinct strips of films with dimensions of 12 × 1 mm^2^ that were ~1.7 um thick and had different Al-Cu compositions were deposited abreast on a 12 × 25 mm^2^ Si substrate. The Cu-alloying concentrations in the strips were chosen to cover the whole range from 0 at% to 100 at%, thus providing a comprehensive study of the whole AlCu layer system. The micro-combinatorial method allowed for the preparation of all 15 layers in one vacuum cycle and in one experiment. It ensured identical sputtering conditions for the strips in the UHV vacuum chamber, e.g., residual and Ar working gas pressures, and with regard to the type and amount of impurities in the system; these are key parameters for reproducibility, as they can significantly affect the properties of a coating [41]. This sample design greatly aided the determination of the system’s mechanical properties via nanoindentation measurements. As a result, Figure 10 shows the evolution of the nano-hardness of the AlCu thin film system as a function of both Cu concentration and max. indentation load. In this layer system, two unexpected phenomena were found: at a concentration of 52 at%Cu, an exceptionally high strength of ~16 GPa was observed, approaching or even reaching the strength of some industrial protective coatings [42].

In addition to high hardness, Figure 10 represents another phenomenon, namely, that at low alloying concentrations, indentations at 10, 20, and 50 mN result in identical H values, while in the intermediate range at ~40–70 at%Cu, lower maximum indentation forces are coupled with higher H values. This interesting phenomenon is called the indentation size effect (ISE), where smaller indentation traces (smaller maximal indentation forces) are associated with higher hardness values for the material. The ISE makes it very difficult to compare the results of hardness measurements taken at different loads. There is a widely accepted theory developed by Nix and Gao [43] that uses geometrically necessary dislocations (GNDs) for the explanation of these phenomena; unfortunately, there is still some uncertainty about its origin, as many factors can have an influence on the determination of hardness, such as the friction between the indenter tip and the indented material [44] or imperfections of the indenter tip [45].

From the comparison of the microstructures of the layers (not shown here) with and without ISE, the otherwise surprising lack of an indentation size effect in the pure Al and pure Cu layers [40] could be attributed to the predominance of the sliding deformation mechanism of the grain boundaries [46].

Thus, it can be concluded that the micro-combinatorial technique has allowed for sampling with a planned, suitable concentration resolution for a comprehensive characterization of binary layer systems. As a result, two phenomena have been discovered and interpreted, namely, the extreme hardness of a binary alloy layer containing rather soft constituents (Al and Cu) and the indentation size effect (ISE).

### 5.4. A-SiGe and Hydrogenated A-SiGe Combinatorial Thin Films

The research of thin, amorphous (a-) SiGe and hydrogenated a-SiGe (a-SiGe:H) films is motivated by their technological importance and wide range of applications in solar cells [47,48,49,50], thin film transistors [51,52], Schottky diodes [53,54], thermal sensors [55], bolometers, etc. SiGe-on-Si is of particular interest because varying the Ge concentration of this alloy allows for the fine tuning of material properties such as bandwidth or the refractive index. In the SiGe system, the operating range can be extended by up to 8 μm (the wavelength of the onset of Si absorption) and further extended up to 14 μm (Ge transparency cut-off wavelength) for devices with a low Si concentration [56,57,58]. 

In this section, it is shown that the micro-combinatorial method has proven to be a very effective technique for the efficient investigation of the structural and optical properties of SiGe and SiGe:H layer systems. This has been demonstrated both in the preparation of a single sample containing a complete binary layer system and by providing optimal conditions for a very efficient characterization of the sample through a range of analytical techniques [59,60] (as shown in Figure 11). While the original procedure [22] of depositing a 1 mm concentration-gradient-length film on a 3 mm diameter grid (Figure 11a) was adequate for TEM studies, to determine optical properties, the sample had to be enlarged to approximately 1 × 0.5 inches, which is in line with the smallest focal spot size (~0.3 mm) typically available for spectroscopic ellipsometry (SE) [61] measurements. Figure 11b shows a schematic of the composition-spread SiGe layer on the 10 × 25 mm^2^ Si substrate, spanning a 20 mm gradient region in the center of the sample. SiGe films were prepared by dual DC sputtering of Si and Ge targets in an Ar atmosphere, while SiGe:H samples were sputtered in a mixture of high-purity H and Ar at p_H_/p = 0, 0.05, 0.10, and 0.20 H partial pressures. For the detailed description of the experimental procedure, see Ref [59,60]. The excellent reliability of the micro-combinatorial method for achieving the desired concentration profile is illustrated in Figure 11d, which shows a very well-matched linear variation of the Si and Ge atomic concentrations along the substrate through two independent methods: Rutherford Backscattering Spectrometry (RBS) and Energy Dispersive X-ray Spectrometry (EDS). The RBS result is indicated by a solid line (red for Si and blue for Ge), and the result of EDS is plotted with a dashed line (only the variation of Si is shown).

Figure 12 shows the atomic fractions of Si, Ge, and H determined by RBS and by elastic recoil detection analysis (ERDA) for layers sputtered in H/Ar mixtures as a function of the sample’s position. Clearly, the hydrogen content of the layer of the hydrogenated SiGe:H samples also varies linearly with the sample’s position, i.e., the concentration of Si [59,60].

In order to reveal the optical properties of the SiGe and SiGe:H systems in the entire composition range, the 25 mm × 10 mm sized µ-combinatorial samples were scanned by an M-2000DI rotating compensator spectroscopic ellipsometer [59,60]. The measuring spot was moved along the center line of the wafer. The plane of incidence was parallel to the short edge, the incidence angle varied between 60° and 70°, and the corresponding size of the focused spot was 0.3 mm wide and 0.6–0.9 mm long. The measuring time was a few seconds for each spot and each angle of incidence. The applied conditions provided high resolution and accuracy of the data library within a reasonable acquisition time of the entire map of the spectra. The detailed measurement parameters and the construction principles of the optical model used for the interpretation of the measured data were published earlier in Ref [59,60].

Figure 13 shows the refractive index (n) and the extinction coefficient (k) values revealed for the amorphous hydrogenated SiGe films prepared at P_H_/P_sputt_ = 0, 0.05, 0.1, and 0.2 H partial pressures as a function of photon energy and sample position. It can be observed that both n and k show a linearly dependent behavior with respect to the sample’s position, i.e., with respect to the Si-Ge composition. These maps are parts of a library of the *optical properties* of the hydrogenated amorphous SiGe binary system covering the entire composition range and a wide hydrogen concentration and wavelength range. The above published library and its numerical data supplements [60] can be used as a reference database for researchers and technologists working with a-SiGe:H.

### 5.5. Characterization of Ternary Metal-Oxynitride (MeON) Layer Systems

Thin layers of metal-oxynitrides (MeON) are widely used (Me denotes Al, Ti, Hf, and Si) for the wavelength-selective coatings of optical elements, smart windows, and laser diodes [62]. Their advantageous feature is their chemical stability and tunable refractive index determined by the proportion of the oxide and nitride components. Clearly, there is a remarkable level of interest in investigating the relationships between sputtering parameters, layer composition, and optical properties in a high-throughput manner over the full range of O/N ratios. However, to prepare composition-spread metal-oxynitride layers via reactive radiofrequency (RF) sputtering, one must regulate the partial pressure of oxygen and nitrogen during the deposition of the combinatorial layer.

The affinity of metals requires the oxygen proportion in the Ar-O-N plasma to be set orders of magnitude lower than that of nitrogen. In one approach, the required low partial pressure range of oxygen was covered by a self-regulating gas inlet assembly based on a peristaltic pump and a finite volume vial [63]. Accordingly, during reactive sputter deposition, while keeping the nitrogen input constant, the oxygen input, and thus the partial pressure of oxygen, gradually decreased toward the chamber base pressure, resulting in a steady decrease in the oxygen content of the deposited layer and thus a gradual transition of the film from oxide to nitride. This experimental set-up facilitates a micro-combinatorial approach to the fabrication and, consequently, testing of three-component layer systems in addition to the two-component systems used so far, which further increases the applicability of the new technique. 

#### 5.5.1. Hf-Oxynitrides

Due to their excellent thermal and chemical stability and corrosion resistance, transition metal nitrides are under intense theoretical and experimental investigation [64,65,66,67,68]. Hafnium nitride systems are used in cutting-edge technologies, for example, as a barrier in a Cu/Hf–N/Si heterostructure [69,70]. An issue related to hafnium nitrides is their extreme sensitivity to a defective structure, especially with respect to non-stoichiometry. On the other hand, Hf oxide is an extremely high dielectric constant material (k = approximately four to six times higher than that of SiO_2_) that is currently used in semiconductor technology for gate dielectrics in metal oxide semiconductor field effect transistors [71]. Ultrathin HfON is a promising candidate for use as a trapping layer for charge-trap memory [72] and as a component of HfMoN(L)/HfON/Al_2_O_3_ tandem absorbers developed for high-temperature solar thermal applications [73].

This emphasizes the need for the comprehensive study of the physicochemical and structural properties of the HfON thin film system as a function of the related fabrication parameters and elemental composition, e.g., the oxygen and nitrogen content. In a one-sample concept micro-combinatorial study, Safran et al. investigated the correlations between the sputtering conditions, composition, and optical properties of the Hf-O-N thin film system over the entire O-N composition range [63].

To this end, the vacuum chamber for RF sputtering was equipped with the previously mentioned O_2_ gas inlet system [63], whose schematic is shown in Figure 14a. The measured and calculated O_2_ partial pressure values as a function of sputtering time and covering the range between 9 × 10^−5^ and 1 × 10^−5^ mbar are shown in Figure 14b.

The RF sputtering of the Hf target took place in an Ar/N/O atmosphere, where the Ar and N pressure was kept constant, and the O partial pressure was continuously decreased. The resulting O and N ratios in the film along the 10 × 25 mm^2^ sized sample were determined by SEM EDS, while the optical properties, including the refractive index (n), were determined by easily applicable spectroscopic ellipsometry. The data obtained are summarized in Figure 15, whose results show a monotonic increase in the refractive index from 2.05 to 2.6 that was observed as O was depleted in the film [63].

#### 5.5.2. Si-Oxynitrides—Modelling Sputtering Parameters

Pure silicon dioxide (SiO_2_) and silicon nitride (SiN_x_) are well-known and essential materials in electronic technology and are produced using a series of production techniques [74,75]. In its complex form, amorphous, ternary silicon oxynitride (SiON) is a chemically stable material with a refractive index between 1.45 and 2.05 depending on the oxygen and nitrogen content. This wide refractive index range would be very useful in optoelectronic applications. However, this layer system is hardly used due to technological difficulties wherein the composition of the SiON and thus its refractive index are particularly sensitive to small changes in sputtering parameters, especially the oxygen content of the plasma gas [62]. Therefore, there is a strong motivation to investigate the sputtering process itself and explore the correlations between the sputtering conditions, composition, and optical properties of the Si-O-N thin film system over the whole composition range.

To achieve this as efficiently as possible, micro-combinatorial layer synthesis was used [76]. A variable layer composition along the substrate was created via the RF sputtering of a Si target with variable reactive gas injection using a method similar to the previously discussed Hf-ON films: In a single sample, an amorphous Si-O-N layer with variable properties was deposited via reactive RF sputtering through a moving slot on a 25 mm long substrate to cover the entire transition from pure Si oxide to Si oxynitride with a variable composition to pure Si nitride. Different target voltages (1.62 kV and 1.95 kV) were used as a further parameter to sputter layers of different thicknesses. The variation of the sample’s optical properties and layer thickness was investigated by spectroscopic ellipsometry (SE) measurements, while the elemental composition was studied by energy dispersive spectrometry (EDS) [76].

Figure 16 shows the remarkable effect of varying the partial pressure of O and N gas injection on the composition of the deposited layer, which varies with the distance measured along the sample, causing a continuous SiO_x_ → SiO_x_N_y_ → SiN_x_ transition. Figure 17 shows that the refractive index (n) of the layer can be tuned in the range of 1.48–1.89 due to the varying partial pressure of oxygen injected into the chamber. Using the data on the composition of the layer, the typical physical parameters of the process were determined using the Berg model for reactive sputtering [77,78]. A new approach was introduced in the modelling procedure: a metallic Si target sputtered with a uniform nitrogen and variable oxygen gas flow was considered an oxygen-gas-sputtered SiN target [76]. According to the calculations detailed in the above publication, the sputtering gas temperature can increase by up to 40 °C during the growth of the oxygen-rich layer due to the exothermic nature of the oxidation process.

A variable reactive gas inlet, the use of a combinatorial layer growth method, and the correlations found between the sputtering parameters, layer composition, and refractive index, which are in agreement with the model detailed in [76], allow for the production of silicon-oxynitride layers with exactly designed optical properties. Moreover, the micro-combinatorial technique facilitates the growth of thin films with the gradient-refractive indices required for some specific applications.

## 6. Conclusions

We have reviewed the combinatorial methods used for the study of multicomponent material systems in chemistry, biology, and materials science with a particular focus on a new micro-combinatorial approach developed for the characterization of multicomponent thin films. This novel method owes its effectiveness to the fact that complete two- and three-component layer systems with variable compositions are prepared and tested within a single sample adapted to the analytical method.

We prove through examples that single-sample concept micro-combinatory methods allow for the high-throughput characterization of the compositionally dependent properties of thin films by a series of measurement techniques, including the measurement of structure and morphology by SEM and TEM; composition by EDS, RBS, and ERDA; mechanical properties by nanoindentation; and optical characteristics by spectroscopic ellipsometry. The examples also highlight the adaptability of the method since, alongside dual DC magnetron sputtering, samples can be produced by RF and reactive RF sputtering, thus extending the range of thin film systems investigated from metals through semiconductors to metal oxynitrides; for example, the concentration-dependent morphology and phases of Al-Mn layers were determined, and the correlations between the structural and mechanical properties—i.e., the hardness and deformation mechanisms—of AlMg and AlCu layer systems were revealed at up to 30%Mg and in the full concentration range, respectively. By using micro-combinatory methods, the optical properties (**n** and **k**) of the complete binary system of amorphous H:SiGe were revealed as a whole material database. The variations in the refractive index (n) of the ternary systems of Me-oxinitrides (Me=Hf, Si) were revealed in the whole composition range from Me-oxide through Me-oxynitride to Me-nitride.

In summary, the new micro-combinatorial method allows for an efficient and comprehensive characterization of the structure and material properties of multicomponent thin film systems, revealing their interrelationships and determining their underlying mechanisms. The scientific results already achieved, as well as the innovation potential of the high-throughput concept, including the creation of two- and three-component thin film databases, can be well exploited for research, technology, and application.

## 7. Patents

The following patent has been granted by the Hungarian Intellectual Property Office for the method and device used to implement a designed variable layer composition change, which allows for the comprehensive investigation of the composition-dependent properties of entire thin film systems: Sáfrán, G. Hung. Patent No. P 15 00500 (2015).

## Figures and Tables

**Figure 1 materials-16-03005-f001:**
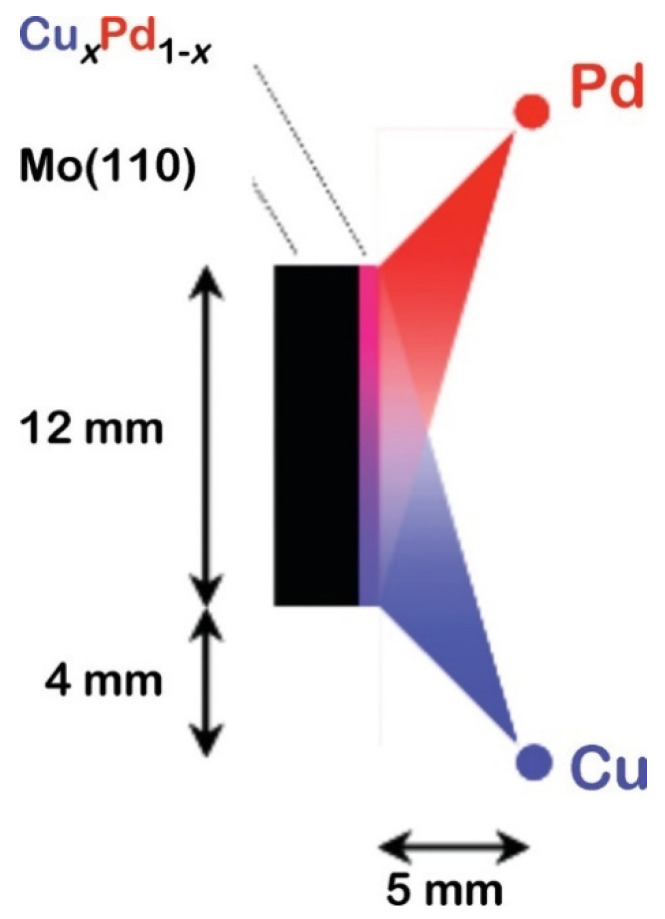
Schematic diagram of the evaporation geometry used for preparing Cu_x_Pd_1-x_ Composition Spread Alloy Films (CSAFs). The Pd and Cu evaporative line sources (normal to the plane of the figure) are offset from the center of the Mo (110) substrate but are close enough to create a gradient in flux across the substrate’s surface. Reproduced with permission from [14]. Copyright © 2011, American Chemical Society.

**Figure 2 materials-16-03005-f002:**
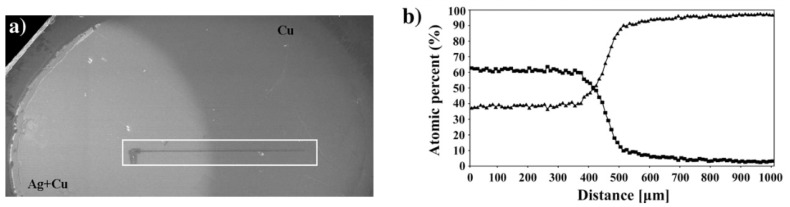
Optical micrograph of surface topography with indentations across the composition gradient in a Cu–Ag nanocomposite (**a**). The variation of the composition is shown in (**b**): ▪ Ag concentration; ▴ Cu concentration. Reproduced with permission from [21]. Copyright © 2007, Elsevier.

**Figure 3 materials-16-03005-f003:**
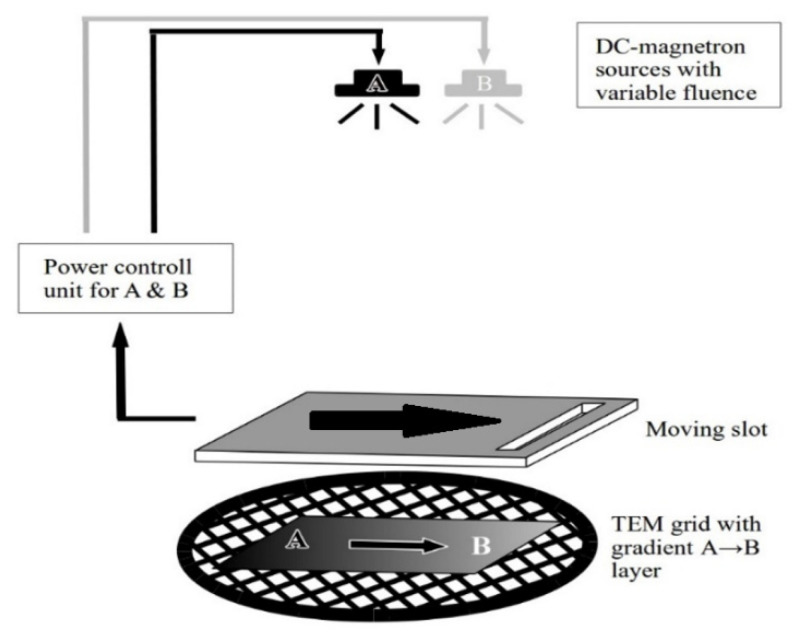
Setup of the synthesis of a micro-combinatorial TEM sample. The concentration-spread layer is deposited from DC magnetron sources A and B through a moving slit onto the TEM grid. The output power of the two sources is adjusted in accordance with the moving slit to obtain a strip of a 3-stage sample: pure A, gradient A ⇨ B, and pure B.

**Figure 4 materials-16-03005-f004:**
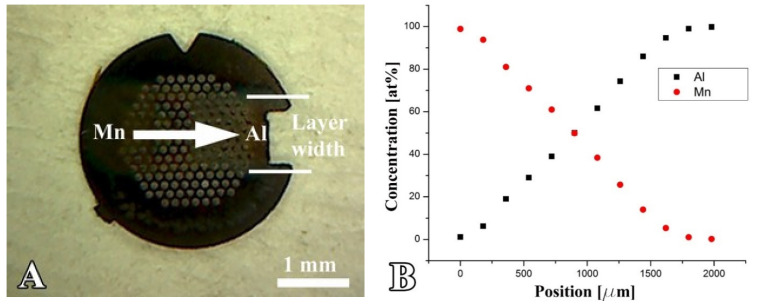
(**A**) Thin Mn–Al micro-combinatorial sample deposited on a Mo TEM grid via dual DC magnetron sputtering. The 1 mm wide layer strip consists of a 1.5 mm long gradient Mn–Al section enclosed by 0.25 mm long pure Mn and pure Al sections. (**B**) Mn and Al concentration profiles measured by EDS as a function of the position along the TEM grid shown in “A”.

**Figure 5 materials-16-03005-f005:**
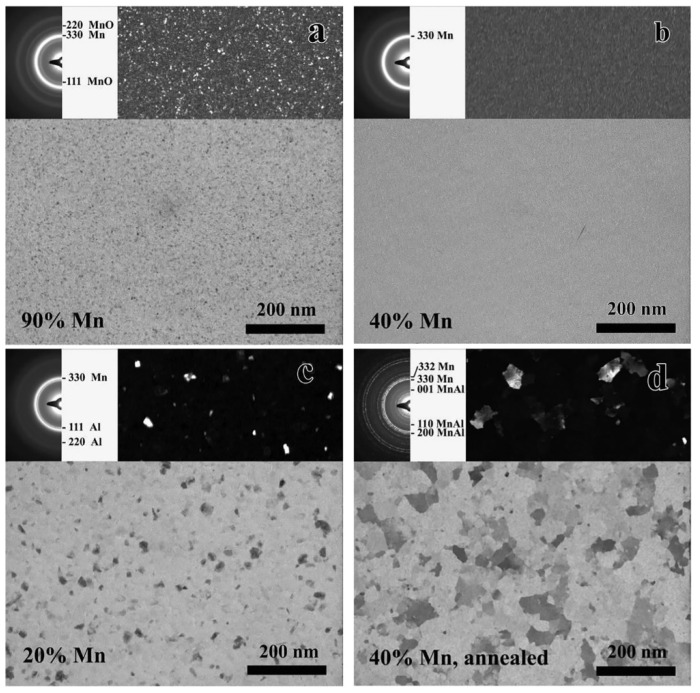
TEM micrographs of a Mn–Al µ-combinatorial sample at (**a**) 90%, (**b**) 40%, and (**c**) 20%Mn concentrations showing fine-grained cubic Mn with Mn oxide, amorphous MnAl, and amorphous MnAl with an fcc Al phase, respectively. The layer depicted in (**d**) is that of 40%Mn after heat treatment at 500 °C/0.5 h in Ar–H_2_ mixture showing tetragonal MnAl phase. The insets are SAED patterns and DF images. Reproduced with permission from [22]. Copyright © 2018, Elsevier B.V.

**Figure 6 materials-16-03005-f006:**
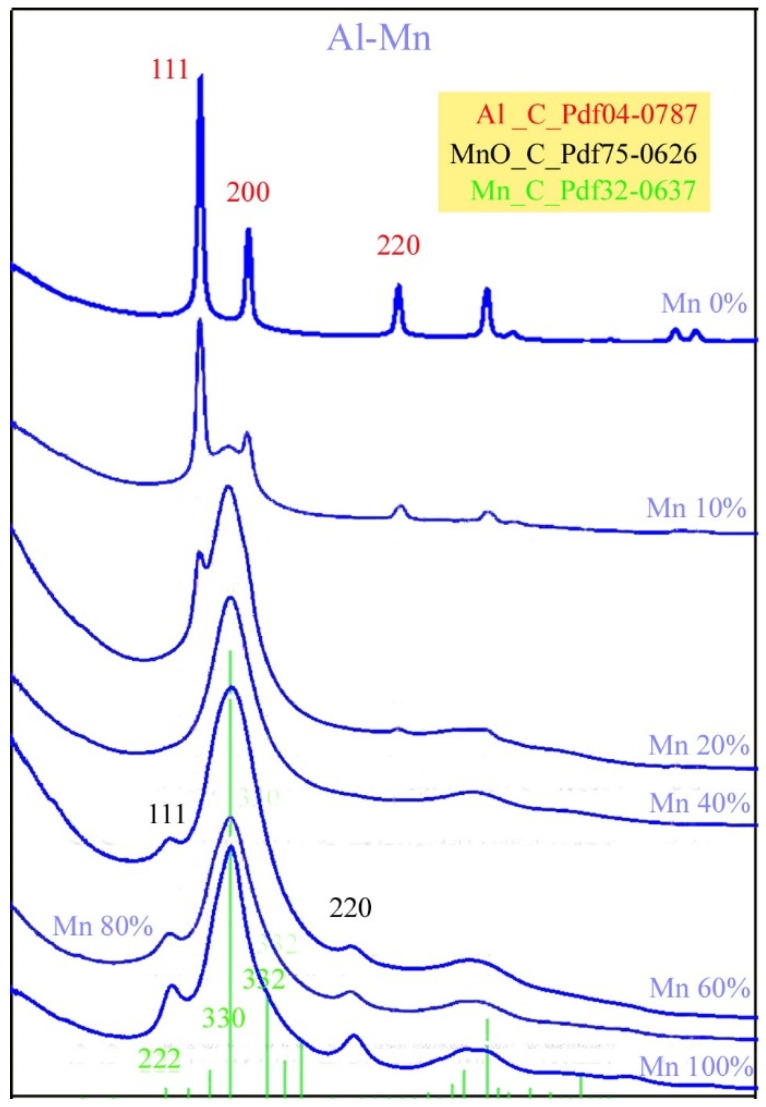
SAED intensity distributions representing the phases of the AlMn layer depicted in Figure 5 with various compositions obtained from a single µ-combinatorial TEM sample deposited at room temperature. Green, black, and red indices show the peak positions of cubic Mn, MnO, and Al phases, respectively.

**Figure 7 materials-16-03005-f007:**
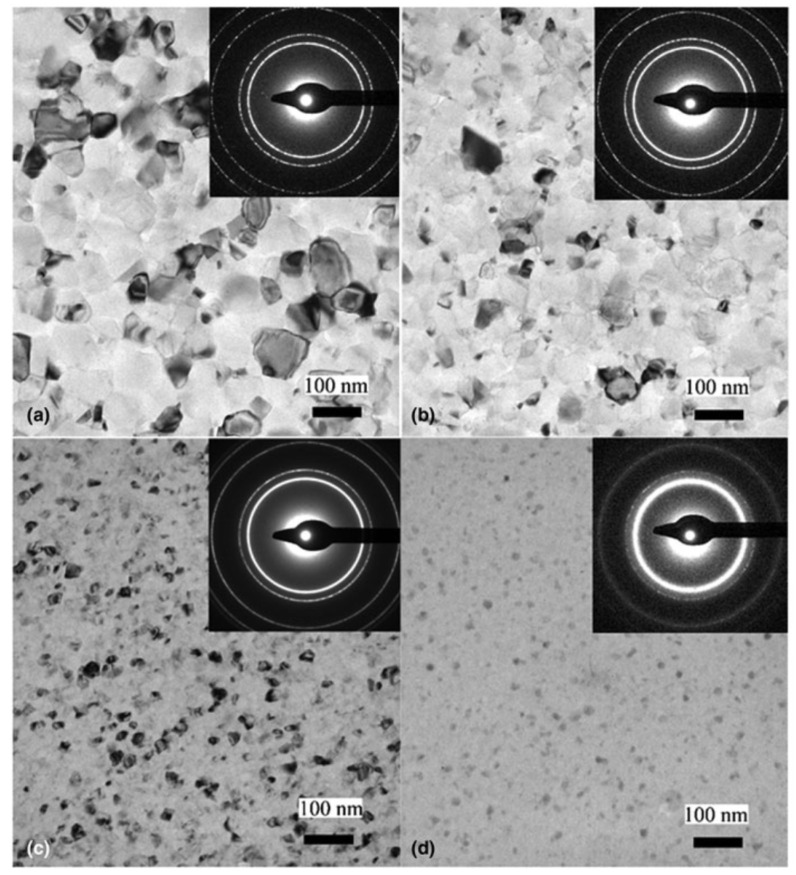
TEM images with SAED insets of (**a**) pure Al, (**b**) Al–1%Mg, (**c**) Al–10%Mg, and (**d**) Al–30%Mg films, showing microstructures with very fine grain sizes decreasing with an increasing Mg content. The films exhibit fcc Al(Mg) phase at each composition up to 30%Mg, at which point an amorphous component becomes dominant. Reproduced with permission from [31]. Copyright © 2019, The Materials Research Society.

**Figure 8 materials-16-03005-f008:**
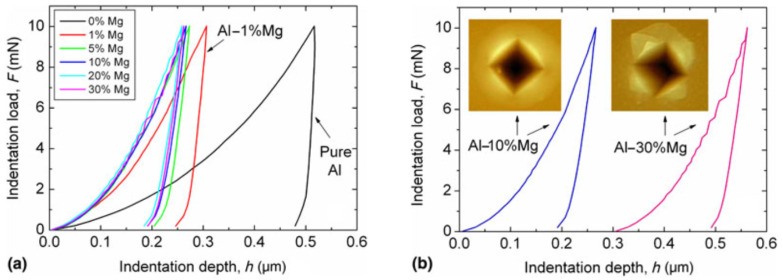
Indentation depth–load (h–F) curves of (**a**) the samples of various Al–Mg compositions and (**b**) of the Al–10%Mg and Al–30%Mg samples together with the corresponding AFM images regarding the indented surface (the h–F curve of the sample Al–30%Mg sample is shifted to the right by 0.5 μm for clarity). Reproduced with permission from [31]. Copyright © 2019, The Materials Research Society.

**Figure 9 materials-16-03005-f009:**
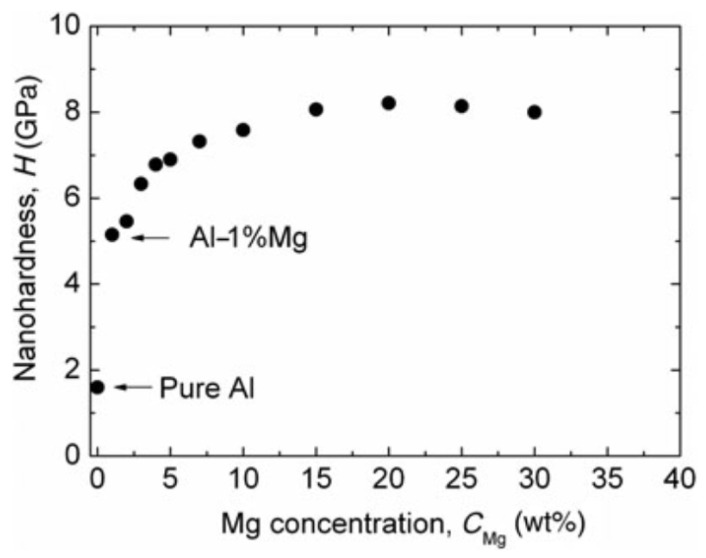
Hardness of the thin AlMg layers as a function of Mg content. Reproduced with permission from [31]. Copyright © 2019, The Materials Research Society.

**Figure 10 materials-16-03005-f010:**
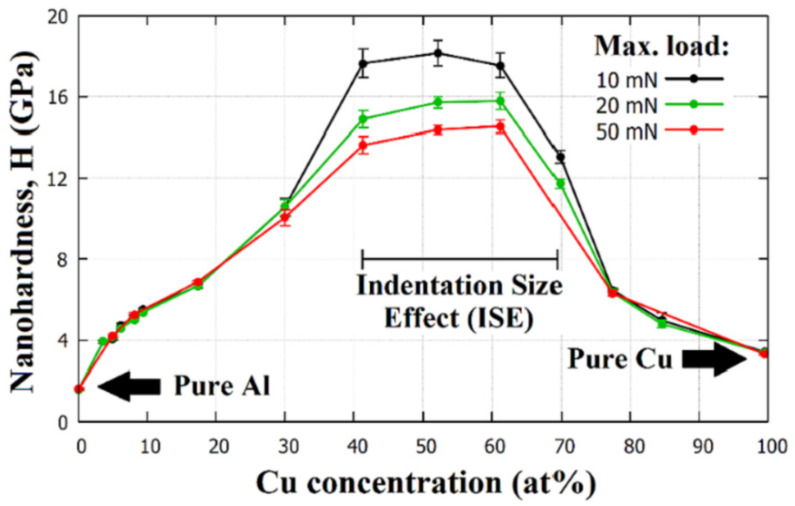
Result of nanoindentation measurements determined from the micro-combinatorial sample: nano-hardness values of the AlCu thin film system as a function of Cu alloying concentration. Reproduced with permission from [40]. Copyright © 2022, Elsevier.

**Figure 11 materials-16-03005-f011:**
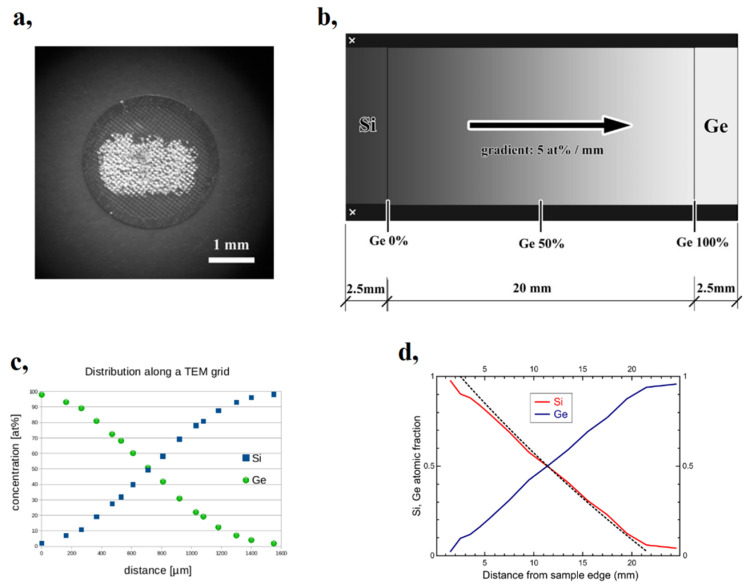
Si_x_Ge_1-x_ (0 ≤ x ≤ 1) thin films fabricated via the micro-combinatorial method (**a**) on a Mo grid for TEM measurements. (**b**) A schematic of the sample for spectroscopic ellipsometry investigations. (**c**) The compositional change across the TEM grid obtained by TEM EDS and (**d**) along the 10 × 25 mm^2^ Si slab obtained via RBS (continuous line) and EDS (dashed line); (**d**) was reproduced with permission from [59]. Copyright © 2018, MDPI, Basel, Switzerland.

**Figure 12 materials-16-03005-f012:**
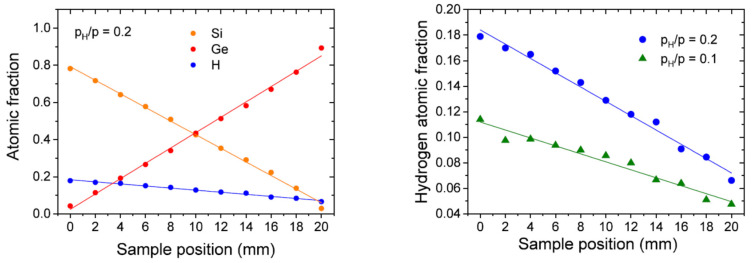
(**Left**): atomic ratios of Si, Ge, and H measured by RBS in a-SiGe:H layer along 20 mm. (**Right**): H atomic fractions of samples sputtered in p_H_/p = 0.1 and 0.2 H partial pressure measured by ERDA. The solid line is a linear fit of the measured values. Reproduced with permission from [60]. Copyright © 2020, Springer Nature.

**Figure 13 materials-16-03005-f013:**
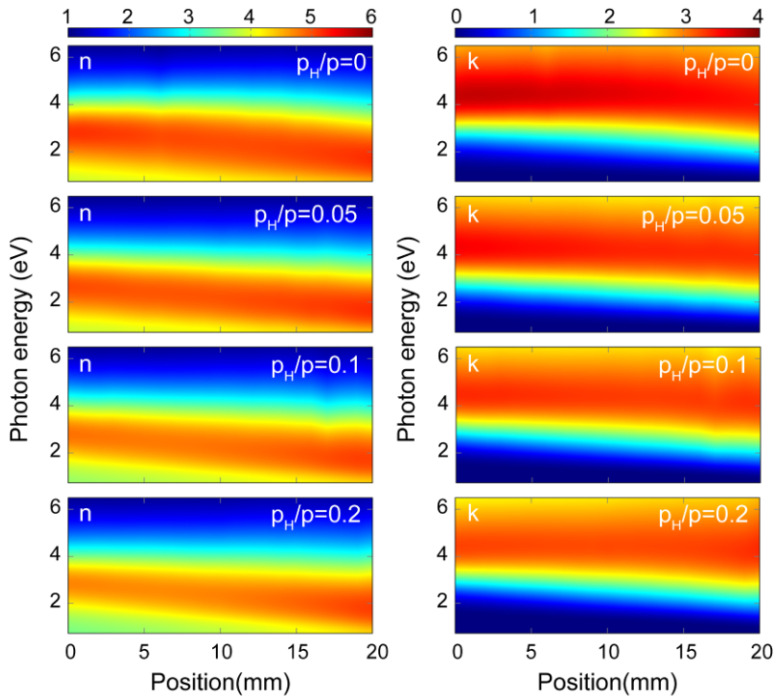
Color-coded maps of refractive index **n** (**left column**) and extinction coefficient **k** of hydrogenated amorphous SiGe films prepared at different H partial pressures (P_H_/P_sputt_ = 0, 0.05, 0.1, and 0.2) as a function of position and photon energy measured by ellipsometry on µ-combinatorial samples. (The 0 mm position corresponds to the pure Si end of the sample.) Reproduced with permission from [60]. Copyright © 2020, Springer Nature.

**Figure 14 materials-16-03005-f014:**
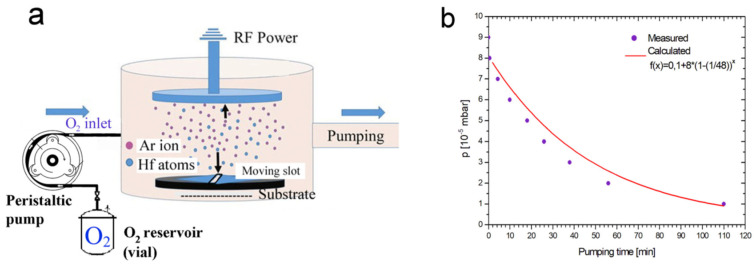
(**a**) Setup of a self-regulating gas injection system that consists of a vial connected to a peristaltic pump. (**b**) Variation in oxygen partial pressure vs. time in the sputtering chamber achieved by the self-regulating gas inlet. The peristaltic pump was operated at a constant rate of 1 sccm emptying the 48 cm^3^ vial of 1 bar of initial pressure. Full circles: measured pressure decay. Solid line: calculated pressure decay. Reproduced with permission from [63]. Copyright © 2020, Elsevier.

**Figure 15 materials-16-03005-f015:**
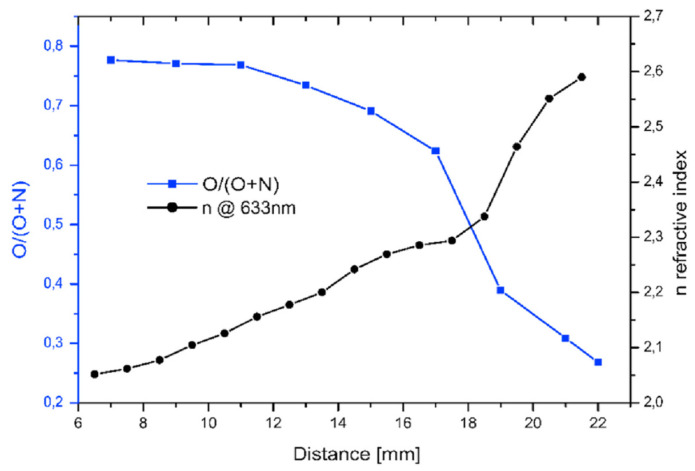
Variation in both oxygen and nitrogen composition (■) and refractive index (_•_) vs. distance along the combinatorial HfON layer RF sputtered in Ar-O-N plasma gas with gradual oxygen depletion. O/O + N ratio revealed by EDS (■); corresponding refractive index revealed by ellipsometry at 632.8 nm wavelength (_•_). Reproduced with permission from [63]. Copyright © 2020, Elsevier.

**Figure 16 materials-16-03005-f016:**
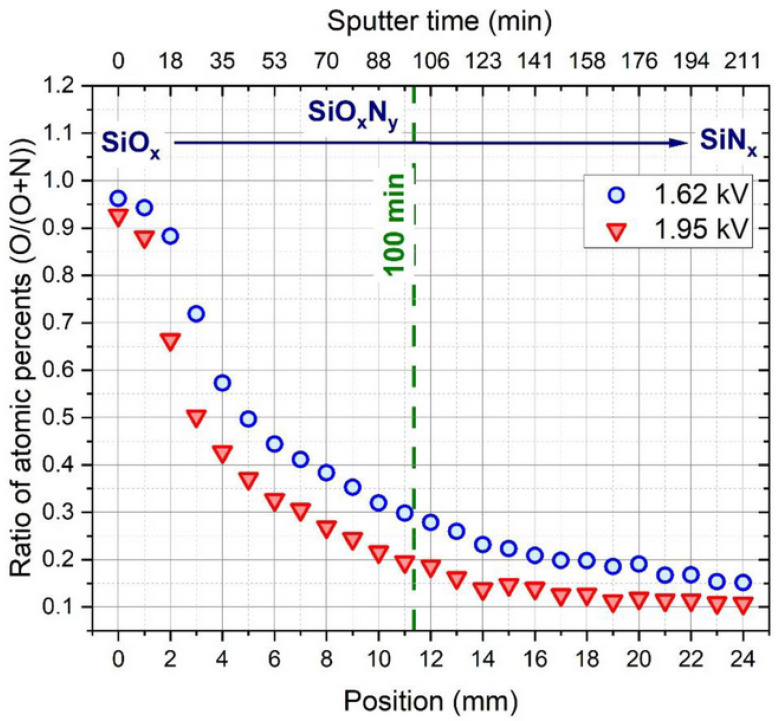
Atomic concentrations (O/O+N) measured by SEM EDS along the micro-combinatorial SiON samples deposited at 1.62 and 1.95 kV. A continuous decrease in the O concentration was observed, resulting in SiO_x_ → SiO_x_N_y_ → SiN_x_ transitions. Reproduced with permission from [76]. Copyright © 2022, MDPI, Basel, Switzerland.

**Figure 17 materials-16-03005-f017:**
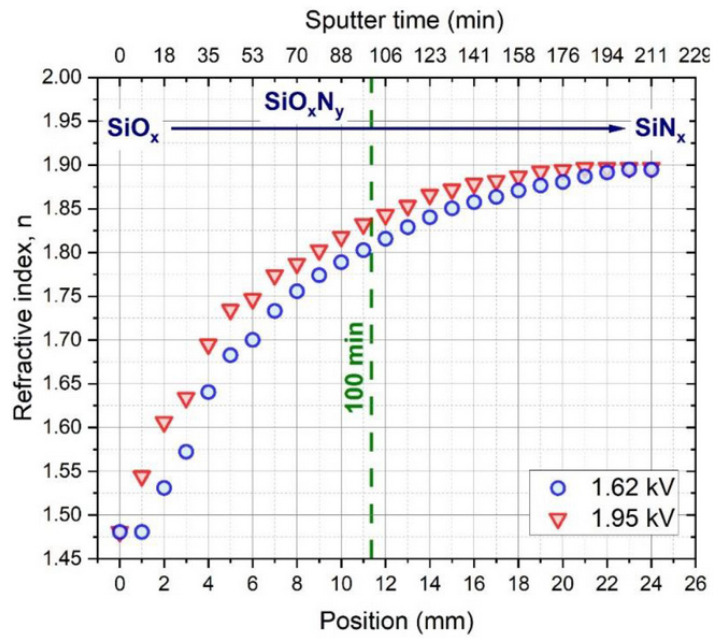
Refractive index (**n**) of samples of varying compositions ranging from 1.48 to 1.89 measured along the substrate by spectroscopic ellipsometry. Reproduced with permission from [76]. Copyright © 2022, MDPI, Basel, Switzerland.

## Data Availability

The data that support the findings of this study were partly taken from the cited references and partly obtained from the results of this paper’s authors, which are not available anywhere else.

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
