# Peer review of "Review on High-Throughput Micro-Combinatorial Characterization of Binary and Ternary Layers towards Databases"

_materials, 2023, doi:10.3390/ma16083005_

Round 1

Reviewer 1 Report

The article covers an interesting study of micro-combinatorial approach to characterization of binary and ternary layers. The work contains both original work as well as review of other works carried out so far in this field. The Authors received a permission from the publishers to reproduce the already published artwork. In my opinion it is well organized and written and supported by appropriate references. The methodology was properly chosen. The results are clearly described and well discussed. The article is of significant importance for this field of research as it embraces a variety of different binary and ternary systems, so, it  may be published in “Materials” journal after minor revision.

The article contains interesting review of micro-combinatorial characterization of binary and ternary layers. The article is well organized and written. The results are clearly described and well discussed. The paper may be published in “Materials” journal after minor revision:

1. Line 132, page 3: “However, it should be noted that the structural characterization of thin films relies primarily on microscopic techniques such as TEM. In light of this, it is indirect and inefficient to deposit a layer of laterally varying composition on a substrate of a few cm, then prepare a series of TEM samples by thinning procedure and TEM-analyze the thinned samples one by one.” -> It is a simplification as cutting-edge SEM microscopes equipped with FEG electron sources could provide an insight into the microstructure of a great deal of PVD/sputtered coatings without a need of cutting out many samples as for TEM studies.

2. Minor language errors (grammar/punctuation) could be corrected i.e. line 88, page 2: "Today’s, integrated circuits, electronics devices, and sensors are based however, mainly, on thin films with widespread physical and chemical properties."

Reviewer 2 Report

This article reviewed the utilization of combinatorial method in the study of multicomponent material systems, which covered a variety range of the materials, properties, and applications. The authors have illustrated the effectiveness of such method comparing with traditional method. According to my opinion, the paper should be published in Materials if the following issues could be addressed.

1.      In figures 3, the authors showed the setup of the sample preparation, where a moving slot was used to deposit the film at different positions. Could the author specify how was the slot movement controlled? If manually, would multi cycle vacuum break (or thermal cycle if substrate was heated) cause some impact on the early deposited film?

2.      As introduced in the experimental setup portion, multi-targets were used for the film composition control. However different targets could have different sputter yield/sputter rate. Beside of the film composition, was there a monitoring of the film thickness?

3.      Usually in sputtering, substrate was heated to achieve better uniformity and roughness, would the diffusion impact the adjacent film areas with different compositions?

4.      In page 12, the authors discussed the indentation size effect (ISE), it would be helpful to add a brief discussion on the origin of such phenomenon.

Reviewer 3 Report

The paper investigates high-throughput and complex characterization of multicomponent thin films over the entire composition range. Combinatorial methods are applicable in different fields of contemporary science i.e. biology, condense physics and chemistry. The authors focus on application of combinatorial methods for characterization of different binary and ternary films prepared by sputtering. Micro-combinatorial technique covers different and well-promising scientific techniques as electronic microscopy, X-ray methods, RBS and AFM. Each of these methods is a very powerful and could be used for individual research. Authors made an interesting assay of completing a mosaic of different techniques together in one paper for different origins of the samples. Micro-combinatorial characterization is an emergency method that is intensively studied nowadays. However, the idea of the article is very risky because one has to be an expert in all fields and all techniques he is referring to, but now all techniques are not fully presented. Authors should moreover be more concentrated on the main idea and practical application of their research to publish this article in “Materials” journal.

The following claims should be also addressed by authors before publishing:  

1.      The name of the article assumes deep description of the combinatorial approach ideas. There is not evident explanation on what it indeed is. Even giving examples describing Dorfman approach does not demonstrate it well enough. Authors are also expected to explain the combinatorial concept more in details before implementing further in the paper.

2.      In abstract different experimental techniques are announced. Afterwards nevertheless only some of them are mentioned in different chapters. Talking on thin films TEM, AFM and XRD are in the focus. However, neither XRD nor AFM experimental are presented in the paper. This is rather strange as there are many different studies performed by AFM and XRD on such objects. Authors cite some of them, but do not show any data. Figure 8 has an inlet with some AFM-images, but these data used just as an illustration to indentation results. Those last are also questionable. It is been mentioned that steeped character of the load curve could only be observed starting 30% Mg sample. Nevertheless, there is no prove of it, as load curves for more than 30% Mg samples are not provided.

3.      Sometimes article language seems to be not scientific: for example mention of “Mother Nature” in the Introduction section.

Approach description should be better optimized for a clear understanding. Maybe authors could use some Tables to compare mentioned methods. 

Reviewer 4 Report

After reading the manuscript I found that the paper is not provide much scientific addition. So I am not recommending for its publication.

Comments:

1.      The introduction is not appropriate to the study.

2.      The research gap is not explained properly.

3.      The discussion on the results is not concrete one rather a descriptive one.

Round 2

Reviewer 3 Report

My comments were not adressed in the response. My opinion stays the same - this articel could be published in someother MDPI journal 

Reviewer 4 Report

The Review article is now in good form. The paper can be accepted.